# Serum Levels of Myonectin Are Lower in Adults with Metabolic Syndrome and Are Negatively Correlated with Android Fat Mass

**DOI:** 10.3390/ijms24086874

**Published:** 2023-04-07

**Authors:** Jorge L. Petro, María Carolina Fragozo-Ramos, Andrés F. Milán, Juan C. Aristizabal, Jaime A. Gallo-Villegas, Juan C. Calderón

**Affiliations:** 1Physiology and Biochemistry Research Group—PHYSIS, Faculty of Medicine, University of Antioquia, Medellín 050010, Colombia; 2Research Group in Physical Activity, Sports and Health Sciences—GICAFS, Universidad de Córdoba, Montería 230002, Colombia; 3Sports Medicine Postgraduate Program, and GRINMADE Research Group, SICOR Center, Faculty of Medicine, University of Antioquia, Medellín 050010, Colombia

**Keywords:** lipids, fatty acids, lipid metabolism disorders, metabolic syndrome, myokine, skeletal muscle

## Abstract

Myonectin has shown beneficial effects on lipid regulation in murine models; therefore, it may have implications in the pathophysiology of metabolic syndrome (MS). We evaluated the relationship between serum myonectin and serum lipids, global and regional fat mass, intramuscular lipid content, and insulin resistance (IR) in adults with metabolic risk factors. This was a cross-sectional study in sedentary adults who were diagnosed with MS or without MS (NMS). Serum myonectin was quantified by enzyme-linked immunosorbent assay, lipid profile by conventional techniques, and free fatty acids (FFA) by gas chromatography. Body composition was assessed by dual-energy X-ray absorptiometry and intramuscular lipid content through proton nuclear magnetic resonance spectroscopy in the right vastus lateralis muscle. IR was estimated with the homeostatic model assessment (HOMA-IR). The MS (*n* = 61) and NMS (*n* = 29) groups were comparable in age (median (interquartile range): 51.0 (46.0–56.0) vs. 53.0 (45.5–57.5) years, *p* > 0.05) and sex (70.5% men vs. 72.4% women). MS subjects had lower serum levels of myonectin than NMS subjects (1.08 (0.87–1.35) vs. 1.09 (0.93–4.05) ng·mL^−1^, *p* < 0.05). Multiple linear regression models adjusted for age, sex, fat mass index and lean mass index showed that serum myonectin was negatively correlated with the android/gynoid fat mass ratio (R^2^ = 0.48, *p* < 0.01), but not with the lipid profile, FFA, intramuscular lipid content or HOMA-IR. In conclusion, serum myonectin is lower in subjects with MS. Myonectin negatively correlates with a component relevant to the pathophysiology of MS, such as the android/gynoid fat mass ratio, but not with other components such as FFA, intramuscular fat or IR.

## 1. Introduction

Skeletal muscle secretes myokines with autocrine, paracrine or endocrine action on multiple physiological processes, such as metabolism, inflammation, and muscle development [1,2,3]. Serum levels of myokines are affected by conditions such as exercise and diet, and likewise, their levels influence the state of health and disease [2,4]. The study of myokines is key to understanding the role of skeletal muscle in the pathophysiology of conditions such as metabolic syndrome (MS) [5].

Central pathophysiological events of MS include alterations in the compartmental distribution of lipids (e.g., increased visceral adipose tissue (VAT)) and insulin resistance (IR) [6]. The increase in VAT leads to alterations in the profile of free fatty acids (FFAs) in serum. For instance, elevations in palmitic, palmitoleic, and dihomo-γ-linolenic acids and low concentrations of linoleic and eicosapentaenoic acids correlate with obesity, IR, increased low-density lipoprotein cholesterol (LDL-C), and total cholesterol (TC) [7,8,9]. In addition, high FFA levels increase ectopic fat in the liver and muscle, which further impairs lipid and glycemic metabolism. Particularly in skeletal muscle, in vivo studies with ^1^H-nuclear magnetic resonance spectroscopy (^1^H-MRS) have reported that intramyocellular (IMCL) and extramyocellular lipids (EMCL) are positively correlated with total fat mass and VAT, IR, and mitochondrial dysfunction [10,11,12].

Several myokines could be protective against metabolic alterations and the risk markers in MS. In this way, irisin, interleukin-6 (IL-6), and apelin improve lipid metabolism and glycemic control [13,14,15]. Similarly, myonectin (erythroferrone or CTRP15) has been proposed as a myokine with an advantageous action on the regulation of lipids, as it reduces circulating levels of total FFA, likely by increasing fatty acid transport in hepatocytes and adipocytes in vitro [16]. In agreement with these findings, myonectin deletion in a murine model increased the accumulation of triacylglycerides (TGs) in white adipose tissue (WAT) while decreasing steatosis in the liver [17]. Additionally, the serum levels of myonectin were reduced in a high-fat diet (HFD)-induced obesity murine model [16].

In human studies, subjects with obesity and type 2 diabetes (T2D) showed lower serum levels than healthy lean subjects, and myonectin levels were inversely associated with indicators of metabolic risk (e.g., high body mass index (BMI), TG, LDL-C, VAT, and IR) [18]. As expected, serum concentrations of myonectin increased in obese women after aerobic exercise training [19]. In contrast, it has also been reported that serum myonectin levels are elevated in obesity, MS, and T2D [20,21,22,23], and a positive association was seen between circulating concentrations of myonectin and fat mass, total FFA, TG, and IR while there was a negative association with high-density lipoprotein cholesterol (HDL-C) [23].

These contradictory findings raise questions about the function and metabolic regulation of myonectin in humans, which deserves additional study. Furthermore, the potential of myonectin as a lipid regulator in humans may be revealed by extending the analysis to the relationship between myonectin and other indicators that are linked to the pathophysiological mechanisms of MS, such as the FFA profile and the intramuscular lipid content. The objective of this study was to evaluate the relationship between serum myonectin and i) lipid components in three different functional and morphological compartments, namely, serum (TG, LDL-C, HDL-C, TC and FFA), body fat depots (global and regional fat mass), and ectopic muscle accumulation (IMCL and EMCL), and ii) IR in adults at metabolic risk.

## 2. Results

### 2.1. Characterization of Study Population

Ninety individuals were included in this study, of whom 61 were diagnosed with MS (Table 1). The groups were comparable in age, sex, and habits (*p* > 0.05), while the values of BMI, fasting glucose, insulin, and HOMA-IR were higher in the MS group (*p* < 0.05).

### 2.2. Results of Serum Lipids, Body Fat Depots and Intramuscular Lipids Content

Regarding lipid outcomes, subjects with MS showed higher TG and lower HDL-C levels (*p* < 0.05). No statistical differences were observed between groups in LDL-C and FFA concentrations (total, saturated, unsaturated, *p* > 0.05). Fat mass index (FMI), VAT, VAT index (VATI) and Android/Gynoid (And/Gyn) ratio were higher in subjects with MS (*p* < 0.05). Intramuscular lipid values (IMCL, EMCL, and total), although higher in MS, did not represent a statistically significant difference (*p* > 0.05) (Table 2).

### 2.3. Circulating Myonectin in Subjects with and without Metabolic Syndrome

Circulating myonectin was lower in subjects with MS than in those without MS (median (interquartile range): 1.08 (0.87–1.35) vs. 1.09 (0.93–4.05) ng·mL^−1^, *p* < 0.05, ES = 0.49), as shown in Figure 1.

### 2.4. Relationship between Myonectin and Lipid Outcomes and Insulin Resistance

We then used a sequential statistical approach to evaluate the relationship between myonectin and the study variables.

For the sake of comparison with previous contradictory literature and because this is the simplest and most used approach to address the relationship between myonectin and lipid outcomes, we first performed a bivariate correlation analysis that showed no significant correlation between myonectin and any study variable (Table 3).

Because some correlation trends (*p* < 0.1) were observed, a multiple linear regression (MLR)-based approach was then performed for all dependent variables shown in Table 3, in order to conduct a more robust statistical analysis. Independent variables were considered as confounders based on biological plausibility and we rigorously verified the assumption that no collinearity exists between them, and residuals had normal distribution.

In a first set of models, adjusted for age and sex, myonectin showed a negative correlation with FMI, VAT, And/Gyn ratio, insulinemia and HOMA-IR (Table 4). Even though myonectin also showed a statistically significant negative correlation with VATI (β = −8.63 g·m^−2^, *p* = 0.012), this was the model with the poorest performance (R^2^ = 0.071, *p* = 0.095).

The models for the above six variables were then further adjusted for BMI (Model I) or indices of fat and lean mass (Model II, Table 5). In both cases, the correlation of myonectin with FMI, VAT, VATI, insulinemia and HOMA-IR disappeared. Interestingly, only the association with the And/Gyn ratio remained significant, and the models became stronger, since the adjusted R^2^ increased from 0.399 in Table 4 to 0.446 in Model II (Table 5). The Durbin–Watson value of 2.0 indicates that no autocorrelation existed in model II, highlighting its reliability. These findings emphasize a negative relationship between myonectin and central obesity.

## 3. Discussion

To shed some light on the potential role of myonectin in systemic and local lipid control in humans, we used a comprehensive approach based on two pillars. First, we studied the correlations between this myokine and lipid variables at different functional and morphological compartments of the human body (i.e., circulation, natural fat depots, ectopic/muscle depots). Second, we moved from the simplest to more complex statistical models when addressing all data from the three above mentioned compartments. This approach allowed us to obtain more reliable results and overcome some limitations found in previous works and which may explain the contradictory results already mentioned. We found that (i) subjects with MS had lower serum concentrations than subjects without MS and (ii) after multiple adjustments, myonectin was negatively correlated only with the And/Gyn ratio.

### 3.1. Subjects with MS Have Lower Serum Concentrations of Myonectin

In this regard, the results reported in the literature are mixed. Mice fed a HFD for 12 weeks showed reduced muscle expression and circulating levels of myonectin when compared to mice fed a low-fat diet [16]. Additionally, lower serum levels of myonectin were observed in subjects with T2D than in controls [18]. Similarly, women with polycystic ovary syndrome (PCOS), a condition that is associated with a variety of metabolic abnormalities, including obesity, IR, and dyslipidemia, have been reported to have lower serum myonectin concentrations than women without PCOS [25]. All of these results are in agreement with our results.

In contrast, Mi et al. [23] reported that subjects with MS have higher values than healthy controls. However, it is noteworthy that subjects from this Asian population had wider age ranges and BMIs (23–82 yr and 15.6–37.59 kg·m^−2^, respectively) than the participants in our study. The study by Park et al. [26] also claimed an increase in myonectin in obese/diabetic animals. However, instead of evaluating myonectin (CTRP15, UniProtKB: Q4G0M1), they evaluated complement C1q tumor necrosis factor-related protein 5 (CTRP5, UniProtKB: Q9BXJ0), which is not the same protein [27]. Finally, other studies with DT2 [20] and PCOS [28] reported that these patients have higher levels of myonectin than controls (-values as mean ± standard deviation- 82.3 ± 47.6 vs. 45.2 ± 23.5 ng·mL^−1^, and -values as mean (Q3-Q1)- 96.3 (50.2) vs. 55.6 (20.4) ng·mL^−1^, respectively).

These contradictory results about circulating myonectin in patients with metabolic disorders may arise due to technical and biological factors, such as the different criteria for patient selection and influence of covariates (e.g., age, level of physical activity, IR, body composition, and geographic region). Furthermore, the time of evolution of the metabolic condition may be associated with differential regulation of myonectin. This situation can lead to phenotypes with different characteristics according to the components and complexity of the MS. For instance, subjects with less VAT, more subcutaneous adipose tissue, and still acceptable lean mass (LM) or subjects with more VAT and less LM are associated with differential lipid and glycemic alterations.

At least three lines of evidence further support our results, i.e., subjects with MS and IR have lower serum myonectin. On the one hand, myonectin levels increase when metabolic conditions improve. For instance, in obese women, myonectin increased after an 8-week aerobic exercise intervention, which was also associated with a decrease in BMI and IR [19]. In this same vein, Li et al. [29] showed that obese patients had a significant increase in myonectin after a gastric sleeve procedure, together with a decrease in BMI and improvements in lipid profile and IR. On the other hand, basic research showed that myonectin increases lipid uptake in the adipose tissue and in the liver, suggesting a causal link between its reduction and an impairment in lipid metabolism. Finally, conditions such as myosteatosis, inflammation, and IR could be linked to a lower expression/secretion of myonectin, as has been reported with other myokines in cases of metabolic disturbances [30,31]. Considering that our participants were diagnosed de novo by our research team, we propose that in the presence of recent, mild metabolic alterations, myonectin levels tend to be decreased. It may be that if the disease progresses, a compensatory increase in myonectin tries to cope with ever-increasing circulating lipids and fat depots taken to the limit.

An important aspect is that myonectin is a regulator of iron homeostasis [32], which suppresses hepcidin transcription by inhibiting the bone morphogenetic protein/SMAD signaling pathway in hepatocytes [33]. Considering that subjects with MS may have iron overload [34] and high levels of serum ferritin and hepcidin [35], it can be hypothesized that low serum myonectin concentrations may be a feedback mechanism to regulate iron overload in this condition, which may explain our results. This is an avenue that should be addressed in future studies.

### 3.2. Correlation of Myonectin with Serum Lipids, Fat Mass, Intramuscular Lipids, and Insulin Resistance

In the simplest statistical model, i.e., a bivariate correlation analysis, no significant relationship was observed with any lipid variable. These results are similar to those reported by Toloza et al. [22] in a rather similar sample. However, significant correlations between myonectin and various metabolic and lipid outcomes have been reported, but in different directions (i.e., negative and positive) with this type of analysis [18,25,28].

To delve into the relationship between myonectin and lipid variables and IR more powerfully, several MLR analyses were performed. According to our analyses, VAT, And/Gyn, and HOMA-IR were negatively correlated with serum myonectin after multiple adjustments. The best regression model (highest adjusted R^2^ and lower D-W value) in our study in humans was for And/Gyn. Notably, these findings are in line with those reported in murine models with Erfe gene deletion, which presented an alteration in the distribution of fat deposits, with increases in white adipose tissue (WAT) and the size of adipocytes and decreased steatosis in the liver [17].

To explain these results, based on in vitro assays, we propose that lower levels of myonectin are linked to less activation of RAC-alpha serine/threonine-protein kinase Akt [36] and a lower lipolytic response of the central adipose tissue, leading to increased fat accumulation. In agreement with this, it was recently reported that gene therapy to overexpress Erfe in HFD-induced obese mice decreased the size of adipocytes, improved the lipolytic response to epinephrine in WAT (the most abundant in the central regions), and also in brown adipose tissue (BAT), further increasing the expression of genes related to thermogenesis (e.g., *Ucp1*, *Pgc-1α*) [37]. Thus, taking together results in murine models and our results in humans, we can propose that under physiological conditions, myonectin modulates lipid turnover through the transport of FFAs, Akt signaling, and the induction of a lipolytic response in central adipose tissue, therefore contributing to an adequate accumulation and distribution of fat. Then, the reduction of myonectin in subjects with MS may partially explain their increased central obesity.

An interesting consequence of these findings is the fact that the relationship between myonectin and IR claimed by previous authors using a simple correlation [18,20,23,25] may not be reliable, since it disappears after adjusting for fat and lean mass. Hence, the relationship between myonectin and IR seems to be mediated by body composition. Alternatively, it may be that the direct effect of myonectin on glucose control is small and can be made visible only under certain conditions. For instance, the overexpression of Erfe in obese mice improved insulin sensitivity and Akt phosphorylation in WAT but not in skeletal muscle [37]. Based on the findings of the previous study, which found no effects on skeletal muscle in mice, it can be proposed that myonectin may have little effect on this tissue in humans, in terms of its lipid metabolism and insulin sensitivity.

On the other hand, despite the reported effect of myonectin on plasma fatty acids in murines [16], we did not find a significant correlation with saturated or unsaturated fatty acids in simple correlations or multiple regression models, which is in agreement with previous works [18,23]. The concentrations of myonectin claimed to have such an effect in murines are between two and three orders of magnitude higher than those usually measured in humans, likely indicating that physiological concentrations of myonectin have a low effect on FFA concentrations in humans.

A few myokines have shown autocrine effects on muscle lipid metabolism: irisin increases oxidative metabolism in the cell line C2C12 [38], apelin increases enzymatic activity and mitochondrial respiratory capacity in mice [31], and IL-6 stimulates intramuscular lipolysis of TG and muscle fatty acid oxidation in healthy physically active men [39]. However, in the case of myonectin, we did not find reports in any model (cell lines, animals or humans) evaluating its autocrine effects on intramuscular lipid content. Considering this gap in knowledge, we measured intramuscular lipids through ^1^H-MRS, which allows a reliable, noninvasive in vivo assessment of IMCL and EMCL lipids, to analyze their relationship with myonectin. Our analyses showed no correlation between myonectin levels and intramuscular lipids. Until we have more information (e.g., the effect of myonectin on fatty acid transporters in muscle), we can conclude that myonectin does not have a measurable effect on intramuscular lipid accumulation.

### 3.3. Strengths and Limitations

We highlight that all measurements were performed with valid and reliable techniques. In the case of ELISA, the proportion of subjects with and without MS was maintained constant in each kit to avoid inter-assay variations that could affect the results. Regarding ^1^H-MRS analyses, we further performed some quality controls by showing that subjects with MS had higher intramuscular lipid concentrations than younger subjects without risk factors (Appendix A), similar to what has been previously reported in the literature [12,40], demonstrating the feasibility of the technique to identify subjects with different amounts of intramuscular lipids. Not all participants had intramuscular lipid measurements; however, the sample of subjects evaluated with intramuscular lipids by ^1^H-MRS remains higher than previous studies with this technique [12,40,41]. Other covariates, such as iron-related markers, were not evaluated. Future studies should consider this variable. Finally, the relationship of some drugs, such as statins, with myonectin is unknown and may influence the correlation between this myokine and clinical outcomes. Nonetheless, analyses performed excluding the group of participants on cholesterol medications showed the same results as those presented in Table 5 (Appendix A), ruling out any bias due to these medications.

To our knowledge, this is the first study to evaluate the relationship between a myokine, specifically myonectin, and lipid variables at different levels of complexity in humans (i.e., serum lipids, total and regional body fat, and intramuscular lipids), which allows a more complete analysis from local to systemic in humans, allowing us to draw stronger conclusions. Additionally, the statistical models were robust, and the comparisons had enough power. Given the scope of this study, the results should be interpreted in the order of the “relationship between variables” and interpretations of causality should be performed with caution.

Finally, our results highlight that robust models should be used to draw reliable conclusions on the role of myokines in humans. Performing several simple correlations among variables increases the probability of false-positives, and authors may reach misleading conclusions.

## 4. Materials and Methods

### 4.1. Study Design

This is a cross-sectional study with individuals of both sexes, aged 40 to 60 years, with metabolic risk, as defined by the presence of at least one diagnostic criterion for MS [42]. Elevated waist circumference was determined according to the cutoff values for the Colombian population (≥92 cm in males and ≥84 cm in females) [43].

Subjects were excluded if they consumed supplements (e.g., vitamin D, creatine or whey protein), received insulin, presented with T2D, had a history of cardiovascular/cerebrovascular events, had cancer, had chronic respiratory disease or were pregnant. Vegetarian/vegan participants or those who followed a ketogenic diet and had some physical or mental limitation were excluded.

The participants underwent a complete medical history, ergospirometry test, biochemical analyses, body composition and intramuscular lipid studies. Subsequently, they were classified into subjects with MS (≥3 MS criteria) or without MS (<3 MS criteria) according to the internationally harmonized criteria of 2009 [42]. Additionally, IR was diagnosed if the homeostatic model assessment (HOMA-IR) value was greater than 2.25 [43].

### 4.2. Ethics Approval

All procedures were approved by the Research Ethics Committee of the Faculty of Medicine at the University of Antioquia (minutes 005 of 15 April 2021) in Medellín (Colombia), in accordance with Resolution number 8430 by the National Ministry of Health of Colombia issued in 1993 and the Ethical Principles for Medical Research Involving Human Subjects outlined in the Declaration of Helsinki in 1975. All participants gave written informed consent.

### 4.3. Experimental Procedures

#### 4.3.1. Complete Medical History and Physical Examination

The clinical history of each participant was recorded, including nutritional aspects and socioeconomic conditions. Physical activity was recorded with the Global Physical Activity Questionnaire (GPAQ) [44]. Cardiorespiratory fitness was assessed through an incremental treadmill ergospirometry test with an open circuit spirometer Oxycon Delta by Jaeger^®^ (VIASYS Health care GmbH, Pullach, Germany), as previously described [45]. Waist circumference was measured with fiberglass anthropometric tape at the midpoint between the lower edge of the rib cage and the iliac crest, according to the World Health Organization recommendations [46], and IR was estimated with the HOMA-IR [5].

#### 4.3.2. Biochemical Analyses

Fasting venous blood samples were collected using serum separator tubes (BD Vacutainer 367815, USA) and centrifuged at 1300× *g* for 15 min at room temperature (Rotofix 32, Hettich, Kirchlengern, Germany). Serum aliquots were kept at −80 °C until processing for myonectin and lipids.

For myonectin measurements, frozen serum aliquots were thawed with no extra heat, and 40 µL was poured into the wells of a human myonectin enzyme-linked immunosorbent assay (ELISA) plate (MyBioSource, MBS1600042, San Diego, CA, USA). This kit has a detection range of 0.05–10 ng·mL^−1^ and a sensitivity of 0.03 ng·mL^−1^ and showed an intra-assay coefficient of variation below 4.0% in our laboratory. All plates were balanced with samples of MS and NMS subjects and were processed by a well-trained investigator blinded to the group coding, according to the supplier’s instructions. Optical density was determined on a plate reader (Varioskan Lux, Thermo Scientific, Waltham, MA, USA) at 450 nm.

The lipid profile (TG, LDL-C, HDL-C and TC) was determined by enzymatic analysis using a Dimension^®^ equipment RXL Max (Siemens, Erlangen, Germany). The FFA measurements were performed by lipid extraction following Folch’s method [47]. For this, 40 µL of tridecanoic acid 50 mg∙mL^−1^ and 2 mL of chloroform:methanol (2:1) were added to 100 µL of serum. After saturation with NaCl, sera were centrifuged at 3400 rpm for 7 min, and the organic phase was separated and dried in a bath at 90 °C. The dry extract was dissolved in 1 mL of hexane and deposited on aminopropyl columns (200 mg). The separation of the FFA was performed using an Agilent gas chromatograph (7890B, Santa Clara, CA, USA) with a flame ionization detector and a TR-CN100 capillary column (60 m × 250 µm × 0.20 µm) [48]. Helium was used as the carrier gas at a flow rate of 1.4 mL·min^−1^. For the identification of fatty acids, the retention times of the samples were compared with a standard (FAME Mix of 37 components, Restek, Bellefonte, PA, USA). Here, we report the cis configuration of the most abundant FFA as detected by the system used: palmitic acid (*n* = 88), stearic acid (*n* = 79), oleic acid (*n* = 73), linoleic acid (*n* = 83), arachidonic acid (*n* = 19), and nervonic acid (*n* = 34).

#### 4.3.3. Body Composition

Global and regional body composition was assessed using dual-energy X-ray absorptiometry (DXA) with the Discovery Wi DXA system^®^ (Hologic, Marlborough, MA, USA) and the Hologic APEX v4.5.3 software (Hologic, USA). All participants were examined in the morning under fasting conditions and a good hydration status (urinary density <1025). We analyzed total fat mass (i.e., fat mass percentage (FM%) and fat mass index (FMI), kg/height(m)^2^); VAT (cm^2^) and VAT index (VATI, g∙m^−2^); and the And/Gyn fat mass ratio, as they are well-known predictors of cardiometabolic risk [49,50]. LM indicators were total lean mass (i.e., lean mass percentage (TLM%), lean mass index (LMI), kg∙m^−2^) and appendicular lean mass index (ALMI, kg∙m^−2^).

#### 4.3.4. Quantification of Intramuscular Lipids

The intramuscular lipid content was evaluated in the right vastus lateralis muscle (VLM) due to its association with the development of metabolic diseases in humans [51,52], using ^1^H-MRS and a flexible coil [45], following international recommendations [53].

Spectra were acquired on a Magnetom Skyra magnet with a Flex large 4-channels, 3T receive only, 516 × 224 mm, coil interface and the SyngoMR D13 program (Siemens Healthcare, Erlangen, Germany) as previously reported [5,45,54]. The spectra were processed with jMRUI [55] v5.2 software (http://www.jmrui.eu, accessed on 1 February 2020). The chemical changes of the protons of the resonances of IMCL and EMCL in the muscle were referenced to the signal of the proton of the water (4.7 parts per million, ppm). Then, Gaussian apodization and subtraction of signals other than methylenes (CH_2_) from IMCL (1.3 ppm) and EMCL (1.5 ppm) were performed on the water-suppressed spectra with the Hankel Lanczos Squares Singular Values Decomposition (HLSVD) method. The quantification of the CH_2_ signals of IMCL and EMCL was carried out with the Advanced Method for Accurate, Robust, and Efficient Spectral fitting (AMARES). Water amplitude and width were analyzed in spectra obtained without water suppression. The relaxation times (T_1_ and T_2_) of IMCL-CH_2_ and EMCL-CH_2_ were taken from Valaparla et al. [56]. The absolute molar concentration of intramuscular lipids per kilograms of wet weight (mmol·kg^−1^ ww) was estimated using a validated equation [57,58]:(1)$$\mathrm{IML}=\frac{\mathrm{ZW}\;\times 10^{6}}{885.4\mathrm{DT}\left(\mathrm{ZW}+P\right)}$$
where IML is the intramuscular lipid content (mmol·kg^−1^ ww.); Z, methylene-to-water spectral intensity ratio (signals obtained from the spectrum of each patient); W, relative tissue water content to total weight of muscle tissue (0.76); T, weighted density of the TG fatty acids relative to the triolein standard, and converted to molar mass by dividing by the triolein standard molecular weight (885.4); D, density of lean muscle tissue (1.05 kg·L^−1^); and *p*, relative methylene proton density of tissue fat versus water (0.61) [58]. The reliability of the quantification of the spectra was high for both IMCL (*n* = 33, r: 0.99 (0.98–0.99), difference in test-retest: 0.40 ± 0.76 mmol·kg^−1^ ww, *p* > 0.05; intraclass correlation coefficient, ICC: 0.99 (0.98–0.99)) and EMCL (*n* = 40, r: 0.98 )0.97–0.99), difference in test-retest: 0.21 ± 3.60 mmol·kg^−1^ ww, *p* > 0.05; ICC: 0.98 (0.97–0.99)). In this study, a total of 49 subjects (MS = 36, NMS = 13) had their spectra analyzed for the quantification of intramuscular lipids.

Figure 2 summarizes the study design and the evaluation of lipid variables at the three different levels of complexity (serum, body fat depots, and muscle).

### 4.4. Statistical Analyses

The sample size was calculated with G*Power v.3.1.9.7, assuming an effect size of 0.71 (difference of myonectin between the groups, MS vs. NMS) [23], an α of 0.05, a power of 85%, and an allocation ratio of 2. We needed to enroll 27 subjects without MS and 55 with MS.

The normality and homoscedasticity of the data were evaluated with the Shapiro–Wilk and Levene tests, respectively. Continuous variables were reported as the mean and standard deviation or median and interquartile range. Categorical variables were expressed as percentages. The comparison between the groups (MS vs. NMS) was performed with Student’s t-test or the Mann–Whitney U test. The comparison of serum myonectin was performed based on the effect size (ES, Hedges’ g) with the novel estimation statistics approach, which allows a complementary analysis to increase the precision and quantitative reasoning of the variables in the context of the study [24]. Chi-square or Fisher’s tests were used for the differences between categorical variables.

Bivariate correlations were performed with Spearman’s Rho (ρ) test (2-tailed). MLR models were performed adjusting for (i) age and sex, (ii) age, sex, and BMI, and (iii) age, sex, FMI, and LMI. A *p* < 0.05 was considered as significant in all analyses. Statistical procedures were conducted with SPSS software v.26 (IBM Corp., Armonk, NY, USA) and R v.4.1.0 [59].

## 5. Conclusions

Myonectin serum levels were lower in participants with MS and negatively correlated with central obesity in subjects with metabolic risk factors. We propose that myonectin is a muscle-secreted factor whose low concentrations may play a role in the pathophysiology of MS, by favoring the accumulation of abdominal fat and the consequent development of IR.

## Figures and Tables

**Figure 1 ijms-24-06874-f001:**
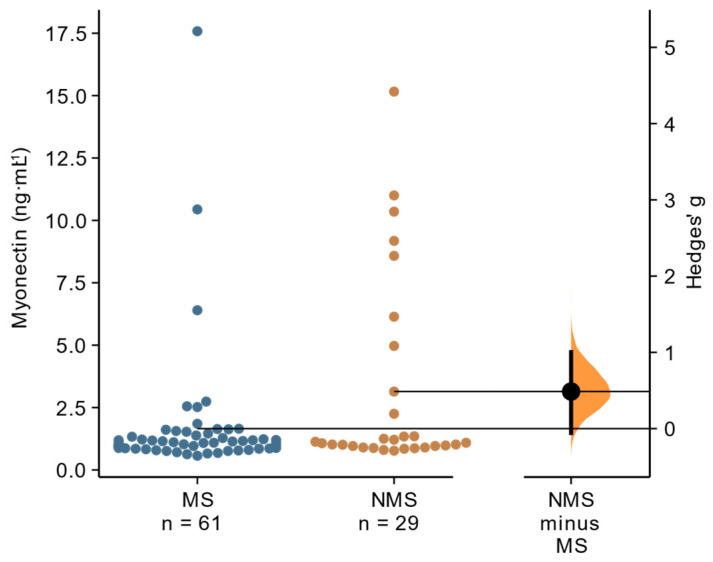
Comparison of serum myonectin concentrations between subjects with MS (metabolic syndrome, blue dots) and without metabolic syndrome (NMS, orange dots). The Hedges’ g between MS and NMS is shown in the above Gardner-Altman estimation plot. Both groups are plotted on the left axes; the mean difference is plotted on a floating axis on the right as a bootstrap sampling distribution (yellow). The mean difference is depicted as a dot; the 95% confidence interval is indicated by the ends of the vertical error bar [24]. The unpaired Hedges’ g between MS and NMS: 0.49 (95.0% CI −0.06–1.0; *p* = 0.026).

**Figure 2 ijms-24-06874-f002:**
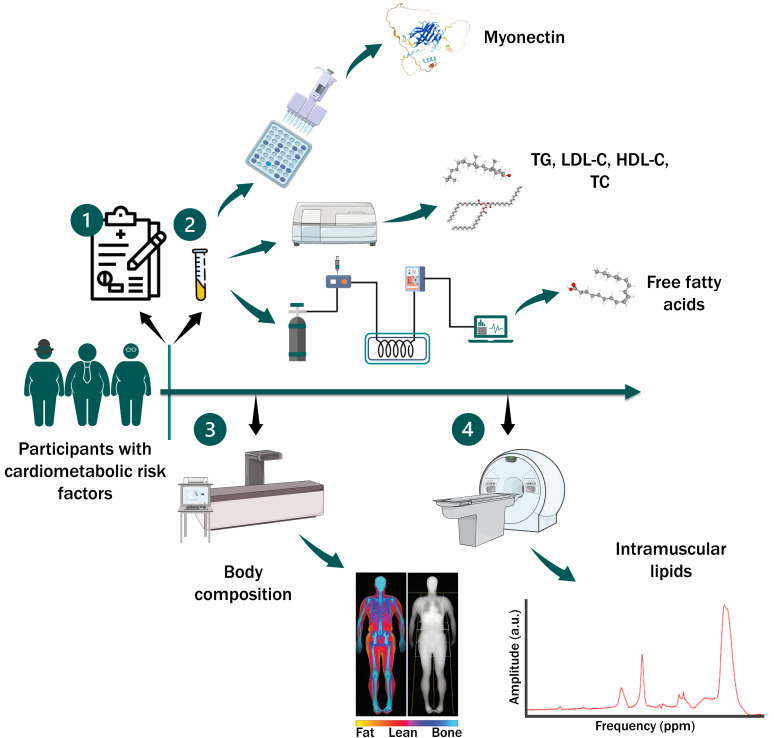
General design of the study, and the evaluation of lipid variables at different levels of complexity. Participants of both sexes, 40–60 years old, were enrolled and evaluated within two weeks. Numbers indicate the experimental procedures: (1) complete medical history, (2) venous blood samples for myonectin, lipid profile and free fatty acids measurements, (3) DXA for global and regional body composition analyses, and (4) ^1^H-MRS for intramuscular lipid content assessment. 3D structures of the presented molecules: myonectin (Uniprot: Q4G0M1; AlphaFold: AF-Q4G0M1-F1); triolein (PubChem CID: 5497163, modeled with UCSF Chimera, v. 1.16); cholesterol (PubChem CID: 5997); linoleic acid (PubChem CID: 5280450). TG, triglycerides; LDL-C, low-density lipoprotein; HDL-C, high-density lipoprotein; TC, total cholesterol; DXA, dual-energy X-ray absorptiometry.

**Table 1 ijms-24-06874-t001:** Characteristics of the study population.

	MS (*n* = 61)	NMS (*n* = 29)	*p*-Value ^a^	ES ^b^
Demographic				
Age, years	51.0 (46.0–56.0)	53.0 (45.5–57.5)	0.572	−0.13
Female, *n* (%)	43 (70.1)	21 (72.4)	0.851	
Anthropometry and lean mass				
Body mass index, kg∙m^−2^	30.6 ± 4.0	27.0 ± 3.5	<0.001	0.94
Waist-to-height ratio	0.61 ± 0.60	0.55 ± 0.06	<0.001	0.89
Waist circumference, cm	94.2 (90.5–102.5)	88.0 (76.1–97.5)	<0.001	0.86
TLM, %	56.7 (53.4–61.5)	58.3 (53.0–66.4)	0.543	−0.17
LMI, kg∙m^−2^	17.3 (15.4–18.9)	15.2 (13.7–17.8)	0.003	0.69
ALMI, kg∙m^−2^	7.4 (6.4–8.1)	6.4 (5.6–7.9)	0.009	0.54
Cardiovascular				
SBP, mmHg	120.5 ± 13.8	119.4 ± 15.7	0.747	0.07
DBP, mmHg	76.7 ± 10.0	72.4 ± 9.5	0.057	0.44
VO_2max_, mL·kg^−1^·min^−1^	29.2 ± 6.3	31.0 ± 5.8	0.247	0.26
Glycemic control				
Fasting glucose, mg∙dL^−1^	97.5 (92.7–105.6)	88.6 (84.4–97.3)	0.002	0.72
Fasting insulin, mIU·L^−1^	14.5 (11.4–19.3)	7.0 (5.1–8.2)	<0.001	1.88
HOMA-IR	3.5 (2.9–4.6)	1.7 (1.1–2.0)	<0.001	1.71
Habits				
Physical activity, min·week^−1^	480.0 (120.0–1020.0)	260.0 (24.0–1960.0)	0.338	−0.21
Smoking, *n* (%)	4 (6.6)	3 (10.3)	0.677	
Alcohol intake, *n* (%)	10 (16.4)	2 (6.9)	0.324	
Medications				
ACEI or ARA II, *n* (%)	20 (32.8)	3 (10.3)	0.023	
Beta-blockers, *n* (%)	6 (9.8)	0 (0.0)	0.080	
CCB, *n* (%)	2 (3.3)	0 (0.0)	0.324	
Diuretics, *n* (%)	11 (18.0)	1 (3.4)	0.094	
Aspirins, *n* (%)	3 (4.9)	0 (0.0)	0.224	
Cholesterol medications, *n* (%)	10 (16.4)	0 (0.0)	0.027	
Metformin, *n* (%)	2 (3.3)	0 (0.0)	0.324	
TG medications, *n* (%)	3 (4.9)	0 (0.0)	0.548	
Anti-HTN medications, *n* (%)	22 (36.1)	3 (10.3)	0.011	

Data are expressed as mean ± standard deviation, median (interquartile range) or number (%); MS, metabolic syndrome; NMS, no metabolic syndrome; TLM, total lean mass; LMI, lean mass index; ALMI, appendicular lean mass index; SBP, systolic blood pressure; DBP, diastolic blood pressure; VO_2max_, maximal oxygen consumption; HOMA-IR, homeostatic model assessment for insulin resistance; ACEI, angiotensin converting enzyme inhibitor; ARA II, angiotensin II receptor blocker; CCB, calcium channel blockers; TG, triglycerides; HTN, Hypertensive. ^a^
*p*-value: Student’s t-test or Mann–Whitney U test (continuous variables), Chi-square or Fisher test (categorical variables); ^b^ ES, effect size calculated with Hedges’ g.

**Table 2 ijms-24-06874-t002:** Results of serum lipids, body fat depots, and intramuscular lipids content.

	MS (*n* = 61)	NMS (*n* = 29)	*p*-Value ^a^	ES ^b^
Serum lipids				
TG, mg∙dL^−1^	182.7 (136.8–219.9)	106.9 (86.0–132.5)	<0.001	0.94
LDL-C, mg∙dL^−1^	138.5 (108.5–183.2)	147.0 (119.9–167.7)	0.852	0.13
HDL-C, mg∙dL^−1^	44.7 ± 10.3	50.9 ± 11.6	0.011	−0.58
TC, mg∙dL^−1^	218.8 (189.7–266.6)	216.2 (193.2–235.8)	0.437	0.31
Palmitic acid, µg∙dL^−1^	0.160 (0.134–0.239)	0.152 (0.116–0.175)	0.846	0.14
Stearic acid, µg∙dL^−1^	0.077 (0.062–0.124)	0.077 (0.054–0.115)	0.191	−0.12
Oleic acid, µg∙dL^−1^	0.081 (0.055–0.124)	0.067 (0.052–0.085)	0.543	0.10
Linoleic acid, µg∙dL^−1^	0.105 (0.081–0.169)	0.097 (0.080–0.124)	0.579	−0.25
Araquidonic acid, µg∙dL^−1^	0.050 ± 0.033	0.042 ± 0.021	0.606	0.26
Nervonic acid, µg∙dL^−1^	0.122 (0.063–0.176)	0.065 (0.050–0.138)	0.140	0.67
Total FFA, µg∙dL^−1^	0.477 (0.350–0.763)	0.395 (0.334–0.504)	0.380	−0.15
Fat mass				
FM, %	40.8 (35.8–44.1)	38.3 (30.4–43.4)	0.400	0.22
FMI, kg·m^−2^	11.6 (10.5–13.7)	9.6 (8.0–11.7)	0.003	0.65
VAT, cm^2^	163.0 (124–204)	128 (92.3–145.5)	<0.001	0.95
VATI, g·m^−2^	316.1 (232.6–391.6)	253.6 (181.9–285.0)	<0.001	0.97
Android/Gynoid ratio	0.58 (0.48–0.72)	0.50 (0.40–0.64)	0.039	0.50
Intramuscular lipids				
IMCL, mmol∙kg^−1^ ww	10.2 (5.6–15.5)	8.4 (5.7–13.5)	0.839	0.19
EMCL, mmol∙kg^−1^ ww	30.0 (21.5–41.6)	25.1 (17.1–44.6)	0.571	0.19
TIML, mmol∙kg^−1^ ww	41.5 (27.6–60.0)	36.2 (26.9–60.1)	0.667	0.20

Data are expressed as mean ± standard or median (interquartile range); MS, metabolic syndrome; NMS, no metabolic syndrome; TG, triglycerides; LDL-C, low-density lipoprotein; HDL-C, high-density lipoprotein; TC, total cholesterol; FFA, free fatty acids; FM, fat mass; FMI, fat mass index; VAT, visceral adipose tissue; VATI, visceral adipose tissue index; IMCL, intramyocellular lipid content; EMCL, extramyocellular lipid content; TIML, total intramuscular lipid content (IMCL + EMCL). Note that the median value of TIML is not expected to be equal to the algebraic sum of medians of IMCL + EMCL. ^a^
*p*-value: Student’s t-test or Mann–Whitney; ^b^ ES, effect size calculated with Hedges’ g.

**Table 3 ijms-24-06874-t003:** Correlations between serum myonectin and serum lipids, body composition and intramuscular lipids content variables.

	ρ	95% CI ^a^	*p*-Value
Serum lipids			
TG, mg∙dL^−1^	−0.01	−0.22–0.21	0.941
LDL-C, mg∙dL^−1^	−0.20	−0.40–0.02	0.062
HDL-C, mg∙dL^−1^	−0.11	−0.31–0.11	0.305
TC, mg∙dL^−1^	−0.19	−0.39–0.02	0.069
Palmitic acid, µg∙dL^−1^	−0.02	−0.24–0.19	0.826
Stearic acid, µg∙dL^−1^	−0.02	−0.25–0.20	0.832
Oleic acid, µg∙dL^−1^	−0.08	−0.31–0.16	0.524
Linoleic acid, µg∙dL^−1^	−0.09	−0.31–0.13	0.399
Arachidonic, µg∙dL^−1^	−0.22	−0.62–0.27	0.367
Nervonic acid, µg∙dL^−1^	−0.31	−0.59–0.05	0.078
Total fat free acids, µg∙dL^−1^	−0.17	−0.37–0.04	0.105
Body composition			
Body mass index, kg·m^−2^	−0.10	−0.31–0.11	0.336
FM, %	−0.15	−0.35–0.07	0.167
FMI, kg·m^−2^	−0.18	−0.37–0.04	0.097
VAT, cm^2^	−0.08	−0.29–0.14	0.463
VATI, g·m^−2^	−0.14	−0.34–0.08	0.192
Android/Gynoid ratio	0.02	−0.20–0.23	0.863
TLM, (%)	0.14	−0.08–0.34	0.198
LMI, kg·m^−2^	0.04	−0.17–0.26	0.676
ALMI, kg·m^−2^	0.06	−0.16–0.26	0.606
Intramuscular lipids			
IMCL, mmol∙kg^−1^ ww	0.10	−0.20–0.38	0.497
EMCL, mmol∙kg^−1^ ww	−0.09	−0.37–0.21	0.547
TIML, mmol∙kg^−1^ ww	<0.01	−0.29–0.29	0.990
Glycemic control			
Fasting glucose, mg∙dL^−1^	−0.15	−0.35–0.07	0.167
Fasting insulin, mIU·L^−1^	−0.14	−0.34–0.08	0.188
HOMA-IR	−0.15	−0.35–0.06	0.155

^a^ 95% confidence intervals (two-sided); TG, triglycerides; LDL-C, low-density lipoprotein; HDL-C, high-density lipoprotein; TC, total cholesterol; FM, fat mass; FMI, fat mass index; VAT, visceral adipose tissue; VATI, visceral adipose tissue index; TLM, total lean mass; LMI, total lean mass index; ALMI, appendicular lean mass index; IMCL, intramyocellular lipid content; EMCL, extramyocellular lipid content; TIML, total intramuscular lipid content (IMCL + EMCL); HOMA-IR, homeostatic model assessment for insulin resistance.

**Table 4 ijms-24-06874-t004:** Multiple linear regression models showing the relationship between myonectin and fat mass depots and insulin resistance.

	Β ^a^	T	*p*-Value	95% CI ^b^	R^2^	Adj. R^2^	*p*-Value	D-W
FMI, kg·m^−2^								
Constant	14.777	6.324	<0.001	10.132 to 19.422	0.344	0.321	<0.001	1.851
Myonectin, ng·mL^−1^	−0.168	−1.918	0.058	−0.343 to 0.006				
Age, years	−0.107	−2.489	0.015	−0.193 to −0.022				
Female sex	3.077	5.167	<0.001	1.893 to 4.261				
VAT, cm^2^								
Constant	201.538	4.241	<0.001	107.060 to 296.016	0.118	0.087	0.013	1.548
Myonectin, ng·mL^−1^	−3.896	−2.184	0.032	−7.442 to −0.350				
Age, years	−0.271	−0.310	0.758	−2.014 to 1.471				
Female sex	−36.557	−3.018	0.003	−60.638 to −12.476				
VATI, g·m^−2^								
Constant	338.342	3.761	<0.001	159.483 to 517.200	0.071	0.039	0.095	1.574
Myonectin, ng·mL^−1^	−8.631	−2.556	0.012	−15.344 to −1.917				
Age, years	−0.277	−0.167	0.868	−3.576 to 3.023				
Female sex	−16.341	−0.713	0.478	−61.930 to 29.247				
And/Gyn ratio								
Constant	0.588	4.641	<0.001	0.336 to 0.839	0.419	0.399	<0.001	1.831
Myonectin, ng·mL^−1^	−0.012	−2.506	0.014	−0.021 to −0.002				
Age, years	0.004	1.601	0.113	−0.001 to 0.008				
Female sex	−0.238	−7.382	<0.001	−0.302 to −0.174				
Insulin, mIU·L^−1^								
Constant	27.393	4.671	<0.001	15.735 to 39.052	0.113	0.082	0.016	1.129
Myonectin, ng·mL^−1^	−0.543	−2.466	0.016	−0.980 to −0.105				
Age, years	−0.218	−2.013	0.047	−0.433 to −0.003				
Female sex	−3.077	−2.059	0.043	−6.049 to −0.106				
HOMA-IR								
Constant	6.359	4.012	<0.001	3.208 to 9.51	0.107	0.076	0.020	1.232
Myonectin, ng·mL^−1^	−0.150	−2.514	0.014	−0.268 to −0.031				
Age, years	−0.044	−1.514	0.134	−0.102 to 0.014				
Female sex	−0.902	−2.233	0.028	−1.705 to −0.099				

^a^ Non-standardized coefficient (indicates the average change that corresponds to the dependent variable for each unit of change in the independent variable); ^b^ 95% confidence interval for β; D-W, Durbin–Watson test; FMI, fat mass index; VAT, visceral adipose tissue; VATI, visceral adipose tissue index; And/Gyn, Android/gynoid fat mass ratio; HOMA-IR, homeostatic model assessment for insulin resistance.

**Table 5 ijms-24-06874-t005:** Multiple linear regression model showing the relationship between myonectin and And/Gyn fat mass ratio.

	Β ^a^	T	*p*-Value	95% CI ^b^	R^2^	Adj. R^2^	*p*-Value	D-W
And/Gyn								
Model I								
Constant	0.259	1.330	0.187	−0.128 to 0.647	0.450	0.424	<0.001	1.954
Myonectin, ng·mL^−1^	−0.010	−2.059	0.043	−0.019 to 0.000				
Age, years	−0.233	−7.362	0.000	−0.296 to −0.170				
Female sex	0.006	2.270	0.026	0.001 to 0.010				
BMI, kg·m^−2^	0.008	2.181	0.032	0.001 to 0.015				
Model II								
Constant	0.483	2.139	0.035	0.034 to 0.932	0.477	0.446	<0.001	2.051
Myonectin, ng·mL^−1^	−0.009	−2.008	0.048	−0.019 to 0.00				
Age, years	0.005	2.105	0.038	0.00 to 0.010				
Female sex	−0.325	−5.721	<0.001	−0.437 to −0.212				
FMI, kg·m^−2^	0.019	2.905	0.005	0.006 to 0.033				
LMI, kg·m^−2^	−0.007	−0.807	0.422	0.034 to 0.932				

^a^ Non-standardized coefficient (indicates the average change that corresponds to the dependent variable for each unit of change in the independent variable); ^b^ 95% confidence interval for β; D-W, Durbin–Watson test; And/Gyn, Android/Gynoid fat mass ratio; BMI, body mass index; FMI, fat mass index; LMI, total lean mass index.

## Data Availability

The data that support the findings of this study are available upon request from the corresponding author.

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
