# Peer review of "Serum Levels of Myonectin Are Lower in Adults with Metabolic Syndrome and Are Negatively Correlated with Android Fat Mass"

_ijms, 2023, doi:10.3390/ijms24086874_

Round 1

Reviewer 1 Report

Dear colleagues

The following manuscript titled by “Serum levels of myonectin are lower in adults with

metabolic syndrome and are negatively correlated with android fat mass”. The paper is well written with novel data.

Myonectin is a lipid regulator in humans that linked to numerous pathophysiological mechanisms including metabolic syndrome. The study reported that patients with MS have lower serum levels of myonectin.  

Author Response

We appreciate the time spent reviewing our manuscript.

We attach a document with the point-by-point response to the reviewers.

Reviewer 2 Report

This entire paper is based on myonectin levels related to parameters of Metabolic Syndrome. MS subjects had lower serum levels of myonectin than NMS subjects (1.08 (0.87–1.35) vs. 1.09 (0.93–4.05) ng·mL. I suppose there is a statistical test to show that these numbers are different, but I believe that biology trumps statistics. There is no biological difference between these averages.

I am not a statistical expert, so I leave that to them. One reviewer for this paper needs to be a biomathematician.

What I do know is that myonectin levels in individuals needs to be paired with parameters of MS for those individuals. I did not see that approach.

There is a lot of good population data regarding parameters of MS. But this is not the paper to publish those parameters.

Author Response

(The authors gave the same response as above.)

Reviewer 3 Report

Comments to Author:

The research is very good and written in scientific language. It is worthy of publication in the journal
Recommendation: Minor Revision

Abstract:

The conclusions of the abstract are not a true reflection of the outcome of the study. Please, rewrite it.

Introduction:

Line 59: and the risk markers

Line 60: (IL-6), and

Line 74: MS, and T2D

Results:

Line 91: while the values

Table 2. rearrangement

Table 4. rearrangement

Author Response

(The authors gave the same response as above.)

Reviewer 4 Report

Reviewer comments and suggestions

The study evaluated the relationship between serum myonectin and serum lipids, global and regional fat mass, intramuscular lipid content, and insulin resistance (IR) in adults with metabolic risk factors. Sedentary adults who were diagnosed with MS or without MS (NMS). 

The study results include MS (n=61) and NMS (n=29) with sex (70.5% vs. 72.4% women). 

MS subjects had lower serum levels of myonectin than NMS subjects (1.08 (0.87–1.35) vs. 1.09 (0.93–4.05) ng·mL-1 P<0.05). Multiple linear regression models adjusted for age, sex, fat mass index, and lean mass index showed that serum myonectin was negatively correlated with the android/gynoid fat mass ratio (R2=0.48, P<0.01).

Below are the comments for this paper to be incorporated in the revised version of the manuscript. 

  1. Line 74 how the authors describe these results.
  2. Line 90-91 Have the authors checked the power analysis of this study
  3. It would be nice if the authors could add ROC CURVE for predicting the values of myonectin in subjects with MS and NMS
  4. Line 160-161 what does it mean?
  5. Line 214-215 Please explain the results.
  6. Line 264 what type of animal model is it?
  7. Line 288-290 what does the author want to say here

Author Response

(The authors gave the same response as above.)

Round 2

Reviewer 2 Report

This reviewer accepts the statistical expertise of the authors and has no further recommendations. Nice paper.